# Does age of ADHD medication initiation predict long-term risk of anxiety? A scoping review

**Margaret Fletcher**[1]*, **Leila Ledbetter**[2], **Patricia Alonso**[1], **Osborn Owusu Ansah**[1], **Olivia Short**[1], **Karin Reuter-Rice**[1]

**1** School of Nursing, Duke University, Durham, North Carolina, United States of America, **2** Medical Center Library & Archives, School of Medicine, Duke University, Durham, North Carolina, United States of America

* margaret.fletcher@duke.edu

## Abstract

Attention deficit hyperactivity disorder (ADHD) is associated with mental health comorbidities, including anxiety. The purpose of this review is to describe evidence regarding the relationship between timing of ADHD medication initiation and long-term anxiety outcomes. Anxiety has a unique relationship to ADHD, as it tends to present earlier in individuals with ADHD compared to those without ADHD and can precede or co-occur with other disorders such as depression. Despite evidence that psychostimulant treatment can reduce short-term anxiety symptoms, the effects of ADHD medication on anxiety long-term are less clear, and the influence of age at medication initiation is unknown. This scoping review included a search of the databases MEDLINE (PubMed), Embase (Elsevier), and Web of Science (Clarivate). The search was conducted by a professional medical librarian in consultation with the author team and included keywords and subject headings representing ADHD, children, medication, and anxiety. Searches yielded a total of 3516 citations after removal of duplicates. Titles and abstracts were screened independently by two reviewers, and conflicts were resolved via discussion. Full-text articles were screened independently by a team of reviewers. Data extraction was completed independently by two reviewers. All screening and data extraction activities were piloted prior to completion. Two articles were selected for inclusion, and neither article found a relationship between age at ADHD medication initiation and long-term anxiety outcomes. Literature examining the relationship between age at ADHD medication initiation and long-term anxiety outcomes is scarce. Neither of the two articles included in this review revealed a significant relationship, however these results must be considered in light of major limitations. Future research is needed to discover the effects of age at ADHD medication initiation on long-term anxiety. This topic is of importance in development of clinical and policy interventions increasing access to early diagnosis and treatment of ADHD.

**Data Availability Statement:** All necessary data required to replicate this review have been uploaded to the Duke Research Data Repository and can be accessed via the following link: https://

doi.org/10.7924/r49p38q4j. Included are the search strategy, screening decisions, and data extraction tables.

**Funding:** The authors received no specific funding for this work.

**Competing interests:** The authors have declared that no competing interests exist.

## Introduction

Attention deficit hyperactivity disorder (ADHD) is a neurodevelopmental disorder that begins in childhood (prior to the age of 12) and is defined by symptoms of inattention, hyperactivity, and impulsivity which result in impaired functioning in at least two settings [1]. ADHD is associated with a multitude of physical and mental health comorbidities, educational and occupational underachievement, and diminished quality of life [2–13]. These problems can persist into adulthood, resulting in significant economic costs due to increased healthcare needs and lack of productivity [14]. Anxiety disorders include social anxiety disorder, specific phobias, panic disorder, separation anxiety disorder, agoraphobia, and generalized anxiety disorder and much like ADHD, can interfere with functioning at school, at home, and with peers [15]. Lifetime prevalence of any anxiety disorder in adolescents with ADHD is estimated to be 35% [7], compared to 32% of adolescents overall [16]. Children with both ADHD and generalized anxiety disorder (GAD) are also at an increased risk (60.6% for ADHD + GAD and 41.1% for ADHD/no GAD) of having more than one mental health comorbidity [17]. Additionally, anxiety disorders present earlier in children with ADHD compared to those without ADHD [18], and mediate the relationship between ADHD and later depression [19].

Receiving medication treatment for ADHD may also reduce anxiety [20, 21]. Stimulant medications are recommended as first-line treatment for individuals aged six and older, though they can be used in preschool-aged children as well [22], and non-stimulant medications can be used alone or in combination with stimulants, and may be especially effective in children with co-occuring ADHD and anxiety [22, 23]. While individuals who have taken medication during childhood experience fewer long-term comorbidities than those who are medication-naïve [24], the impact of age at initiation of medication treatment on secondary comorbidities such as anxiety has not been well-explored. There is evidence that earlier medication treatment and longer duration of treatment for ADHD can reduce the risk of later substance use disorders [25, 26], but there is less research regarding timing of medication treatment and other mental health comorbidities such as anxiety. One meta-analysis reported a decreased risk in children with ADHD, with higher doses of stimulant medications in randomized control trials (RCTs) corresponding to lower levels of anxiety [20]. However, the RCTs examined only a number of weeks at most and thus do not capture the long-term effect of medication use on development of anxiety symptoms and disorders. Therefore, the objective of this scoping review is to answer the following question:

1. What literature exists comparing age at initiation of ADHD medication to long-term anxiety outcomes?

This scoping review will elucidate the impact that timely medication treatment of ADHD can have on anxiety in children. Through the exploration of existing literature, we can assess the relationship between earlier initiation of ADHD medication and risk of anxiety symptoms or disorders in children with ADHD. This has important implications for clinical practice and policy, potentially highlighting the importance of access to earlier diagnosis and treatment.

## Methods

### Design

This work is a scoping review, carried out using the Johanna Briggs Institute (JBI) Manual for Evidence Synthesis [27] and was reported following the Preferred Reporting Items for Systematic reviews and Meta-Analyses extension for Scoping Reviews (PRISMA-ScR) Checklist (S1 Table). A systematic review protocol was registered through Prospero (PROSPERO 2024 CRD42023447273). However, throughout the review process, it became clear that there is a

dearth of literature in this area and we would not be able to draw conclusions regarding the relationship between age at ADHD medication initiation and long-term anxiety. As such, it was decided that this paper better aligned with the scoping review methodology, with the aim being to map the literature surrounding this topic rather than answer a specific research question [28].

### Information sources

The databases searched included MEDLINE (PubMed), Embase (Elsevier), and Web of Science (Clarivate). Although the JBI Manual for Evidence Synthesis does not specify the number of databases that should be searched, section 10.2.5 states that the search should "aim to be as comprehensive as possible within the constraints of time and resources" and that "any limitations in terms of the breadth and comprehensiveness of the search strategy should be detailed and justified [27]." We selected databases based on the Cochrane Handbook (section 4.3.1) which recommends MEDLINE, Embase and a third subject specific database [29]. Additionally, Bramer et al conclude that MEDLINE, Embase and Web of Science are an optimal combination of databases to achieve adequate coverage and a high recall percentage [30]. The search was developed and conducted by a professional medical librarian in consultation with the author team and included a mix of keywords and subject headings representing ADHD, children, medication, and anxiety. The searches were independently peer reviewed by a librarian using a modified PRESS Checklist. Search hedges or database filters were used to remove publication types such as editorials, letters, comments, adult-only and animal-only studies as was appropriate for each database. In addition, a search hedge was used to limit the studies to cohort and case-series study types in order to narrow the results to reflect appropriate methodologies for this topic. The search was conducted on July 17, 2023, and found 4516 citations. An updated search was conducted on July 10, 2024 and 357 additional articles were uploaded and screened in Covidence, 11 of which were selected for full-text review and of those, none met inclusion criteria for the present scoping review. Complete reproducible search strategies, including date ranges and search filters, for all databases are detailed in S2 Table. Articles that cited or were cited by each article were screened for relevant content by a single author (M.F.), and no further relevant articles were identified.

### Eligibility criteria

Articles were considered for inclusion in this review if they included children and adolescents under age 18 with ADHD and examined age at initiation of medication treatment in relation to long-term anxiety symptoms, disorders, or medication use. Articles were excluded if the sample included co-occurring developmental disorders (e.g. Autism Spectrum Disorder), learning disabilities, underlying neurological conditions or injury, or genetic syndromes. Editorials or commentaries were also excluded as well as previous literature reviews.

### Selection of evidence sources

Following the search procedure, all identified studies were uploaded into Covidence (Veritas Health Innovation, Melbourne, Australia), a software system for managing systematic reviews, and 1,000 duplicates were removed by the software. A final set of 3,516 citations were left to be screened in the title/abstract phase. Four reviewers participated in study selection, with both the lead author and one other reviewer assessing each study independently. M.F. and P.A. completed title/abstract screening, each independently reviewing the articles. Articles were excluded if they clearly did not meet inclusion criteria based on title and/or abstract review. All disagreements were resolved by discussion among reviewers.

Four reviewers (M.F., P.A., O.A., and O.S.) completed the full-text screening stage. Papers were reviewed in detail by two independent reviewers and were excluded if they did not meet the eligibility criteria. Any conflicts between the independent reviewers were resolved through adjudication by a separate reviewer. For papers not published in English that met the inclusion criteria during the title/abstract screening, the abstracts were reviewed for usable data. For papers not published in English that met the inclusion criteria during the title/abstract screening, the abstracts were reviewed for usable data and are presented in S3 Table. Due to limited funding for translation services, accurate data collection from papers in languages other than English was not feasible and they were excluded at the full-text screening phase. The article selection is presented by flowchart as per PRISMA guidelines (Fig 1).

## Data collection process

Data extraction was completed using a template within the Covidence software, which was built and tailored to this study by the lead author. The data extraction template was piloted with all reviewers and updated according to feedback from pilot testing. Two reviewers (M.F. and P.A.) completed data extraction independently and resolved any conflicts through discussion. Data extracted included: general information, study characteristics, measures, statistical methods, findings, and limitations (see Table 1 for a detailed list of data extracted).

## Data items

The main variables under consideration in this review were age at initiation of ADHD medication (any stimulant or non-stimulant used to treat ADHD in children), and any measure of long-term anxiety, which could include diagnoses, symptoms, or anxiety medication use. Long-term was defined as an anxiety outcome measured at least one year after starting ADHD medication. These broadly defined measures were selected since the aim of this review was to determine what, if any, literature exists on this topic with the expectation that this topic had not likely been extensively studied.

## Synthesis methods

Data were synthesized in narrative form, with the goal of answering the research question for this study, which was "What literature exists comparing age at initiation of ADHD medication to long-term anxiety outcomes?" No quantitative syntheses or meta-analyses were performed, and no qualitative articles met inclusion criteria. The data extraction tables were examined in detail by the lead author and summarized in the results section below. The data extraction tables and the narrative results were then examined by all reviewers to prevent any missing data [31].

## Results

After removal of duplicates, a total of 3516 citations were screened at title and abstract. 214 articles were reviewed at full text. Following full-text screening, two articles remained that fit the eligibility criteria for this study. A brief description of each study is provided in Table 2.

The first article, by Madjar et al. [32], evaluated the relationship between childhood methylphenidate (MPH) adherence and prescriptions of antidepressant medications (ADM) during adolescence, which are indicated in the treatment of depression and/or anxiety [33]. This was an observational, longitudinal retrospective study using data from Clalit Health Services, the largest payer-provider system in Israel which provides health services across all Israeli districts. The researchers evaluated the prescription histories of 6,834 children in the Clalit Health

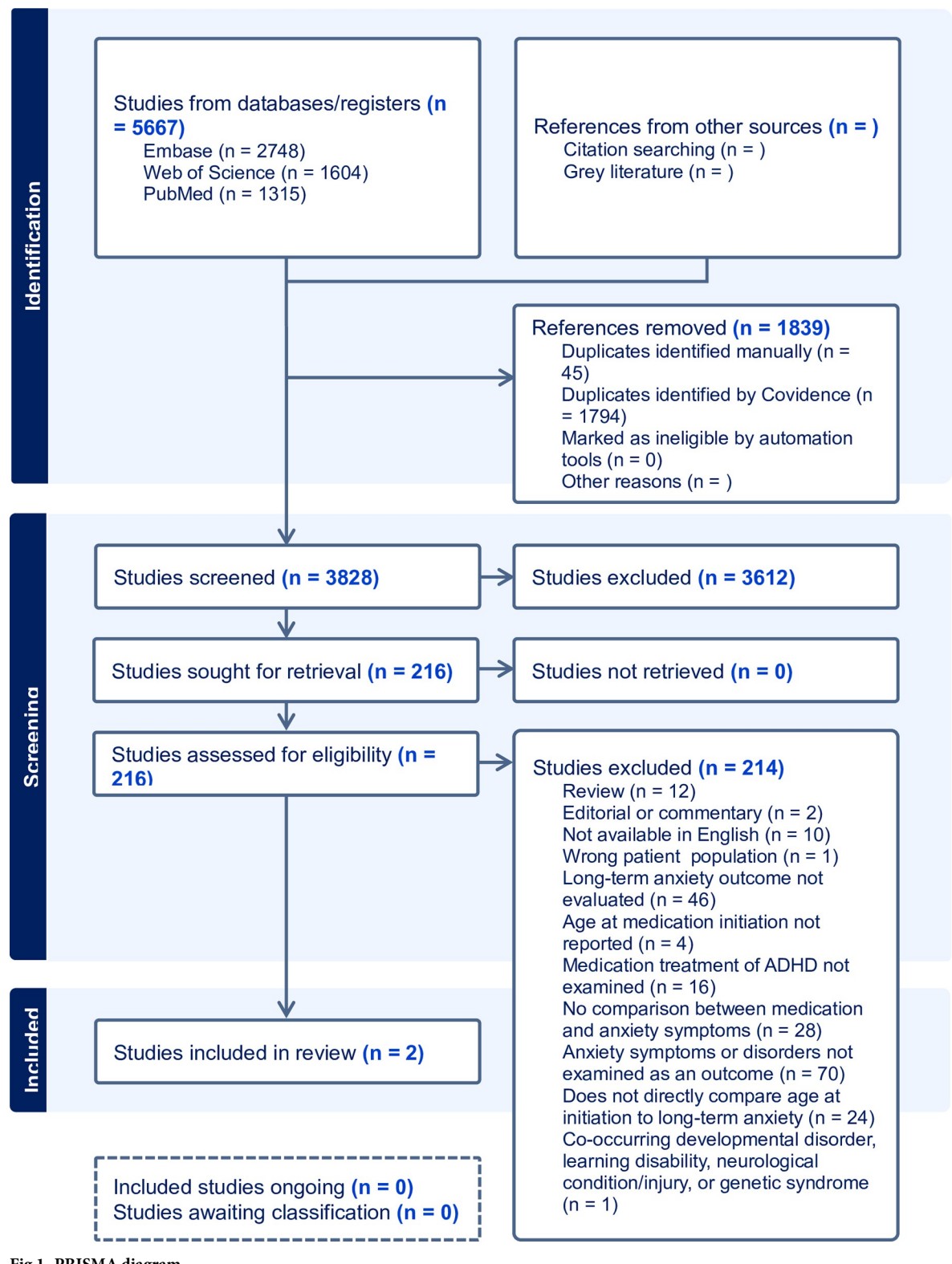

**Fig 1. PRISMA diagram.**

**Table 1. Data extracted.**

| General information |
| --- |
| • Study ID, title, authors, lead author contact details, publication year, journal, country in which study was conducted, notes |

| Study characteristics |
| --- |
| • Methods: aim(s), hypotheses, study design, study design details, start date, end date, funding sources, possible conflicts of interest, inclusion criteria, exclusion criteria, population of interest, method of recruitment, number of participants |
| • Sample characteristics (age, sex, race, socioeconomic status, other) |

| Measures |
| --- |
| • Predictors: how are participants determined to have ADHD, medications evaluated, aspects of medication use evaluated, age at medication initiation, notes |
| • Outcomes: anxiety measures, timing of anxiety measure (compared to ADHD medication initiation), age of anxiety measure |
| • Covariates, other measures, notes |

| Statistical methods |
| --- |
| • Data management |
| • Data analysis |
| • Notes |

| Findings |
| --- |
| • ADHD medication and anxiety, other findings, relevant limitations, notes |

Services database. They hypothesized that greater adherence to MPH prior to the age of 12 would predict lower risk of dispensed ADM after the age of 12. Participants were born between January of 1990 and December of 1996, were first prescribed MPH between January of 1998 and December of 2002 (at ages 6–8), and were not prescribed any ADMs prior to the age of 12.

**Table 2. Overview of studies extracted.**

| Title | Childhood methylphenidate adherence as a predictor of antidepressants use during adolescence | Age of methylphenidate treatment initiation in children with ADHD and later substance abuse: prospective follow-up into adulthood |
| --- | --- | --- |
| Authors | Nir Madjar, Dan Shlosberg, Maya Leventer-Roberts, Amichay Akriv, Adi Ghilai, Moshe Hoshen, Amir Krivoy, Gil Zalsman, Gal Shoval | Salvatore Mannuzza, Ph.D., Rachel G. Klein, Ph.D., Nhan L. Truong, M. A., John L. Moulton III, Ph.D., Erica R. Roizen, B.A., Katheryn H. Howell, B.S., and Francisco X Castellanos, M.D. |
| Publication year | 2019 | 2008 |
| Journal | European Child and Adolescent Psychiatry | American Journal of Psychiatry |
| Country | Israel | United States |
| Design | Retrospective, observational, longitudinal chart review | Prospective, non-randomized longitudinal cohort |
| Data collection dates | January 1998-December 2014 | Recruitment 1970–1977, end date not specified |
| Sample | N = 6,834 Israeli children ages 6–8 prescribed ADHD medication based on evaluation by a certified child psychiatrist | N = 176 U.S. based White boys ages 6–12 years old of middle socioeconomic status referred to a psychiatric research clinic in New York |
| Methods | Compared methylphenidate (MPH) adherence among children from age 6–8 through age 12 to antidepressant (ADM) prescriptions dispensed between ages 12–18 | Referred children were treated with methylphenidate (MPH) and followed longitudinally in late adolescence and in young adulthood. Follow-up assessments included clinician assessment for antisocial personality disorder, ADHD, substance use disorder (SUD), mood, anxiety, and psychotic disorders |
| Results | Higher MPH adherence predicted higher ADM use (OR = 1.50), stratifying by age at initiation did not alter the findings | No significant relationship was found between age at MPH initiation and anxiety disorders in adolescence and adulthood (Wald $\chi2 = 0.40$, $p > .10$) |
| Discussion points | Results may be attributed to increased likelihood that parents who consistently provide their child with MPH or attain positive results from MPH may be more likely to see medication treatment of future anxiety/ depression | Earlier age at initiation did not *increase* risk of future psychiatric disorders and was associated with a decreased risk of SUD. |

Adherence was evaluated as a percentage (i.e. number of MPH monthly purchases dispensed/ number of total months through age 12). Logistic regression was used to assess the association between MPH adherence during childhood and adolescent antidepressant use (defined as prescription dispensed between ages 12–18). A 50% cutoff was used and tested for adequacy. The researchers also performed sensitivity analyses, stratifying the population by age. They found that children with higher adherence were 50% more likely to receive a prescription for ADM than children with lower adherence. Parental ADM or antipsychotic use also predicted ADM use in adolescence. When sensitivity analysis was performed, age of MPH initiation (6, 7, or 8) did not significantly predict likelihood of receiving ADM [32].

The second article that met inclusion criteria for the present study was by Mannuzza et al. [34]. This was a non-randomized, longitudinal, experimental study with a primary aim to evaluate the relationship between age at MPH initiation and later substance use, but the researchers also examined antisocial personality disorder, mood, anxiety, and psychotic disorders as secondary outcomes. This study recruited 176 White male children between six and twelve years old referred to a psychiatric research clinic between 1970 and 1977 for hyperactivity and treated with MPH during childhood. Children were included in the study if they were English-speaking, had a telephone at home, and were referred to the clinic by their school and found to meet Diagnostic and Statistical Manual (DSM) II diagnostic criteria for hyperkinetic disorder (as ADHD was known at the time). Participants were excluded if they had received previous significant stimulant treatment (>10mg/day MPH), had an intelligence quotient (IQ) less than 85, evidence of psychosis or neurological disorders or demonstrated aggressive or other serious antisocial behaviors. Participants were followed longitudinally, with follow-up visits in late adolescence (age 18.4 $\pm$ 1.3 years) and in adulthood (age 25.3 $\pm$ 1.3 years). They were interviewed at follow-up by clinicians who were blind to their childhood status (i.e. diagnoses and medication), assessing for DSM-III-R criteria for ADHD, conduct disorder, substance use disorder (SUD), mood disorders, anxiety disorders, antisocial personality disorder, and psychotic disorders. Predictors were age at MPH initiation, IQ, childhood hyperactivity severity, socioeconomic status (SES) in childhood, and lifetime parent mental disorders. Findings indicated that there was no relationship between age at MPH initiation and anxiety disorders. However, there was a significant positive association with age at medication initiation and antisocial personality disorder, any SUD, non-alcohol SUD, and stimulant SUD. Additionally, the authors examined SES, which was significantly related to non-alcohol SUD in bivariate analyses, but was no longer significant when entered into the model with age at MPH initiation. These findings indicate that later age at MPH initiation is related to increased risk of SUD and antisocial personality disorder but not anxiety, mood, or other DSM-III-R disorders [46].

## Discussion

The purpose of this scoping review was to map the literature assessing the relationship between the timing of ADHD medication initiation in childhood and long-term anxiety outcomes. The relationship between ADHD and anxiety is well-established [7, 21, 35], and there is conflicting literature regarding the influence of medication for ADHD and long-term risk of anxiety. Some studies have found a reduced risk of anxiety in children treated with medication [36, 37], while others have not found difference in long-term anxiety between treatment/non-treatment groups [38–41]. It has been posited that adverse experiences stemming from ADHD symptoms contribute to the development of secondary mental health comorbidities [21], suggesting earlier initiation of ADHD medication could result in fewer adverse experiences and thus decreased risk of comorbidities. However, the present study suggests that anxiety is one such comorbidity that is underexplored despite its importance for quality of life and its

relationship to other comorbidities [17, 19, 42]. After screening 3,614 articles, we have identified only two articles that address this topic, neither of which are an ideal representation of the topic in question.

Neither article found a relationship between age at ADHD medication initiation and long-term anxiety. While this is reassuring in the sense that earlier age of medication initiation does not appear to increase the risk of anxiety, it does not seem to decrease it either, despite evidence that stimulant treatment decreases anxiety in the short-term [20]. However, neither study was designed to address the specific question of age at medication initiation as it relates to long-term anxiety. Additionally, measurements of social drivers of health (SDOH) (e.g. SES, income) are cosigned to minimized roles in the overall narrative of medically treated ADHD and anxiety. Neither article substantially addressed SDOH that could influence access to mental health care, including diagnosis and treatment of ADHD and anxiety disorders. Mannuzza [34] sampled White boys in the United States, thus protective factors (e.g. privilege based on race or sex, parental support, proximity to clinic) could have influenced the risk of experiencing anxiety or other adverse outcomes. Madjar et al. [32] reported that parental ADM or antipsychotic use was related to higher ADM use in adolescents. The authors noted that this could be attributed to either increased genetic risk of depression or anxiety, or greater inclination to seek mental health treatment. Additionally, greater adherence to MPH in childhood also predicted higher ADM use during adolescence. This could also be attributed to greater parental capacity and inclination to seek and continue with medication treatment. SDOH which contribute to treatment access and beliefs surrounding mental health treatment should be considered in future studies, and further longitudinal research is needed in large cohorts of children treated with ADHD medication in more recent years. The Adolescent Brain and Cognitive Development (ABCD) Study® [43] offers opportunities for such research, as this rich database includes almost 12,000 participants recrtuited from 2016–2018 at ages 9–10 followed longitudinally with a multitude of measures, including mental health diagnoses, medication use, SDOH, and survey data regarding functional outcomes [44, 45].

## Limitations

This review is not without limitations. First, although non-English articles were included in the screening process, they were excluded at the point of full-text review due to lack of resources for translation services. As such, relevant articles could have been missed leading to geographical bias. Second, a cohort and case study filter was applied in order to increase the quality of the evidence, however that could have excluded studies inadvertently. Third, the purpose of this study was descriptive, and thus conclusions cannot be drawn regarding the relationship between ADHD medication treatment and long-term risk of anxiety.

There were several limitations in the studies extracted as well. The Madjar et al. [32] article used existing data to explore the relationship between MPH adherence in childhood and ADM use in adolescence. Although the study was well-designed to explore the longitudinal effects of MPH use (including age at initiation), the outcome of ADM use does not differentiate between use of ADM for depression versus anxiety. ADMs, including selective serotonin reuptake inhibitors (SSRIs), are first-line treatments for both types of disorders [46, 47], so the sample likely included both uses. Consequently, the results include participants who were taking ADM for both depression and anxiety, and conclusions cannot be drawn regarding anxiety specifically. This study also only considered MPH but did not consider other stimulant medications or any non-stimulant medications, due to policy in the Israeli health network which at the time of data collection only offered MPH treatment. The researchers acknowledged that their study was unable to capture individuals who sought health care in a private setting so

they could access alternatives to MPH. Another major limitation of this study is that the researchers considered a narrow initial age range, including only children starting ADHD medication between ages six and eight. This limits the variability in age of initiation, and thus any effect of age at initiation may have gone undetected [30].

The Mannuzza et al. [34] study should also be considered in light of significant limitations. Initial recruitment occurred in the 1970's and thus does not reflect modern understanding of ADHD diagnosis and/or treatment. The DSM diagnostic criteria have been updated since initial data collection [48] and new formulations of stimulant and non-stimulant medications are available which demonstrate well-established efficacy and tolerability [49]. Additionally, this study relied on a referred clinical sample without experimental control which was homogenous (included only White males at one U.S. clinic) and thus cannot be generalized to females or children from racial or ethnic minority groups. The authors additionally noted that they were unable to separate age at diagnosis/treatment with stimulant medications specifically. Instead, they suggest that the duration of undiagnosed ADHD in childhood may be the important variable in question, rather than stimulant treatment.

## Conclusion

This scoping review highlights the need for more research on the relationship between age of medication initiation for ADHD and long-term anxiety. The two articles included in this study did not find a relationship between age of initiation and long-term anxiety, but were fraught with significant limitations, such as the use of ADM as an outcome, which does not differentiate between use for depression or for anxiety [32], and the use of a cohort treated in the 1970's, prior to the availability of more extensive choices of ADHD medication [34]. Future studies should examine a wider distribution of age at initiation, and should take a more nuanced approach to medication treatment of ADHD by including multiple types of medication (i.e. stimulant vs. nonstimulant) and by examining trajectories of medication use longitudinally to represent consistency of use over time. Furthermore, the role of SDOH in access to timely diagnosis and treatment of ADHD should be at the forefront of future discussions on the relationship between ADHD treatment and long-term outcomes. Data from recent large, longitudinal cohorts are available for these purposes, such as the ABCD Study® [43], which collects data on medication use, SDOH, and mental health symptoms and diagnoses throughout adolescence. Characterizing the relationship between age at initiation of ADHD medications and long-term mental health outcomes will inform the development of future interventions aimed at providing equitable access to diagnosis and treatment of this highly prevalent and consequential disorder.

## Supporting information

**S1 Table. Preferred Reporting Items for Systematic reviews and Meta-Analyses extension for Scoping Reviews (PRISMA-ScR) checklist.**
(DOCX)

**S2 Table. Search strategy.**
(DOCX)

**S3 Table. Non-English abstract summaries.**
(DOCX)

## Author Contributions

**Conceptualization:** Margaret Fletcher, Leila Ledbetter, Karin Reuter-Rice.

**Data curation:** Leila Ledbetter.

**Formal analysis:** Margaret Fletcher, Patricia Alonso.

**Investigation:** Margaret Fletcher, Leila Ledbetter, Patricia Alonso, Osborn Owusu Ansah, Olivia Short.

**Methodology:** Margaret Fletcher, Leila Ledbetter, Karin Reuter-Rice.

**Project administration:** Margaret Fletcher.

**Resources:** Leila Ledbetter.

**Supervision:** Karin Reuter-Rice.

**Writing – original draft:** Margaret Fletcher, Leila Ledbetter.

**Writing – review & editing:** Margaret Fletcher, Leila Ledbetter, Patricia Alonso, Osborn Owusu Ansah, Olivia Short, Karin Reuter-Rice.

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
