## [Decision Letter · Decision Letter 0]

12 Nov 2024

PMEN-D-24-00395

Does age of ADHD medication initiation predict long-term risk of anxiety? A scoping review

PLOS Mental Health

Dear Dr. Fletcher,

Thank you for submitting your manuscript to PLOS Mental Health. After careful consideration, we feel that it has merit but does not fully meet PLOS Mental Health’s publication criteria as it currently stands. Therefore, we invite you to submit a revised version of the manuscript that addresses the points raised during the review process.

We look forward to receiving your revised manuscript.

Kind regards,

Gellan Karamallah Ramadan Ahmed

Academic Editor

PLOS Mental Health

Journal Requirements:

 1. We note that your Data Availability Statement is currently as follows: "This is a scoping review and no underlying data is available to report. To facilitate replication, our search strategy, including all databases and search terms used, is detailed in the supporting information file (S2 Table)."  Please confirm at this time whether or not your submission contains all raw data required to replicate the results of your study. Authors must share the “minimal data set” for their submission. PLOS defines the minimal data set to consist of the data required to replicate all study findings reported in the article, as well as related metadata and methods (https://journals.plos.org/plosone/s/data-availability#loc-minimal-data-set-definition).  For example, authors should submit the following data:  - The values behind the means, standard deviations and other measures reported; - The values used to build graphs; - The points extracted from images for analysis.  Authors do not need to submit their entire data set if only a portion of the data was used in the reported study.  If your submission does not contain these data, please either upload them as Supporting Information files or deposit them to a stable, public repository and provide us with the relevant URLs, DOIs, or accession numbers. For a list of recommended repositories, please see https://journals.plos.org/plosone/s/recommended-repositories.  If there are ethical or legal restrictions on sharing a de-identified data set, please explain them in detail (e.g., data contain potentially sensitive information, data are owned by a third-party organization, etc.) and who has imposed them (e.g., an ethics committee). Please also provide contact information for a data access committee, ethics committee, or other institutional body to which data requests may be sent. If data are owned by a third party, please indicate how others may request data access. 2. Please provide an Author Summary. This should appear in your manuscript between the Abstract (if applicable) and the Introduction, and should be 150–200 words long. The aim should be to make your findings accessible to a wide audience that includes both scientists and non-scientists. Sample summaries can be found on our website under Submission Guidelines:  https://journals.plos.org/mentalhealth/s/submission-guidelines#loc-parts-of-a-submission 

Additional Editor Comments (if provided):

I have completed my evaluation of your manuscript. The reviewers recommend reconsideration of your manuscript following revision. I invite you to resubmit your manuscript after addressing the comments below.

Reviewers' comments:

Reviewer's Responses to Questions

**Comments to the Author**

1. Does this manuscript meet PLOS Mental Health’s publication criteria? Is the manuscript technically sound, and do the data support the conclusions? The manuscript must describe methodologically and ethically rigorous research with conclusions that are appropriately drawn based on the data presented.

Reviewer #1: Yes

Reviewer #2: Yes

2. Has the statistical analysis been performed appropriately and rigorously?

Reviewer #1: Yes

Reviewer #2: Yes

3. Have the authors made all data underlying the findings in their manuscript fully available (please refer to the Data Availability Statement at the start of the manuscript PDF file)?

Reviewer #1: No

Reviewer #2: Yes

4. Is the manuscript presented in an intelligible fashion and written in standard English?

Reviewer #1: Yes

Reviewer #2: Yes

5. Review Comments to the Author

Reviewer #1: The scoping review is a thoughtful piece as it adheres to the Johanna Briggs Institute methodology and PRISMA guidelines. However, I recommend some improvements. Firstly, while the shift from a systematic review protocol to a scoping review approach is mentioned, it would be insightful to provide a more clearer rationale for why this change was necessary and how it impacted the review process and more to that the "web-link" provided in the method section for the study registration should be removed. This clarification would enhance the transparency of the study design. Secondly, although the search strategy includes key databases, expanding the search to additional databases would have helped to provide more comprehensive overview of the available research, Thus clearly explaining the rationale behind this database limitation will be insightful. And lastly, the exclusion of non-English studies due to resource constraints, as mentioned in the limitations, till represents a significant gap. I recommend explicitly readdressing this limitation and considering ways to mitigate it in future reviews

Reviewer #2: This was a very interesting read, easy to understand and informative. I believe that it is a good foundation for further research into affective anxiety as a result of ADHD.

Line 91 - Additionally, missing n.

Line 173 - PRISMA flow chart - was it intended to be in-text?

Line 181/2 Table 1- SES was not defined. First defined in line 248

Line 305 - can not - should read cannot

Line 332 - additionally noted (add y)

6. PLOS authors have the option to publish the peer review history of their article (what does this mean?). If published, this will include your full peer review and any attached files.

**Do you want your identity to be public for this peer review?** For information about this choice, including consent withdrawal, please see our Privacy Policy.

Reviewer #1: No

Reviewer #2: **Yes: **Tola Awe

---

## [Decision Letter · Decision Letter 1]

26 Dec 2024

Does age of ADHD medication initiation predict long-term risk of anxiety? A scoping review

PMEN-D-24-00395R1

Dear Ms. Fletcher,

We are pleased to inform you that your manuscript 'Does age of ADHD medication initiation predict long-term risk of anxiety? A scoping review' has been provisionally accepted for publication in PLOS Mental Health.

Best regards,

Gellan Karamallah Ramadan Ahmed

Academic Editor

PLOS Mental Health

Reviewer Comments (if any, and for reference):

Reviewer's Responses to Questions

**Comments to the Author**

1. If the authors have adequately addressed your comments raised in a previous round of review and you feel that this manuscript is now acceptable for publication, you may indicate that here to bypass the “Comments to the Author” section, enter your conflict of interest statement in the “Confidential to Editor” section, and submit your "Accept" recommendation.

Reviewer #1: All comments have been addressed

Reviewer #2: All comments have been addressed

2. Does this manuscript meet PLOS Mental Health’s publication criteria? Is the manuscript technically sound, and do the data support the conclusions? The manuscript must describe methodologically and ethically rigorous research with conclusions that are appropriately drawn based on the data presented.

Reviewer #1: Yes

Reviewer #2: Yes

3. Has the statistical analysis been performed appropriately and rigorously?

Reviewer #1: Yes

Reviewer #2: Yes

4. Have the authors made all data underlying the findings in their manuscript fully available (please refer to the Data Availability Statement at the start of the manuscript PDF file)?

Reviewer #1: Yes

Reviewer #2: Yes

5. Is the manuscript presented in an intelligible fashion and written in standard English?

Reviewer #1: Yes

Reviewer #2: Yes

6. Review Comments to the Author

Reviewer #1: the authors should proof read the article again to thoroughly address the issue of language in the write up. I believe this scoping review establishes a call to action research necessity around ADHA especially if funding are available to explore more non English studies.

Reviewer #2: The authors have addressed the issues I raised in my initial review and the revision has improved the manuscript. I recommend accepting the paper.

7. PLOS authors have the option to publish the peer review history of their article (what does this mean?). If published, this will include your full peer review and any attached files.

**Do you want your identity to be public for this peer review?** For information about this choice, including consent withdrawal, please see our Privacy Policy.

Reviewer #1: **Yes: **TEKUH ACHU Kingsley

Reviewer #2: **Yes: **Tola Awe
